# Funneling of Oblique Incident Light through Subwavelength Metallic Slits

**DOI:** 10.3390/nano13010061

**Published:** 2022-12-23

**Authors:** Alex E. Chen, Xue-Qun Xia, Jian-Shiung Hong, Kuan-Ren Chen

**Affiliations:** 1Department of Physics, National Cheng Kung University, Tainan 70101, Taiwan; 2School of Electrical Engineering and Computer Science, The Pennsylvania State University, State College, PA 16802, USA

**Keywords:** plasmonics, nanophotonics, FDTD, funneling effect, oblique incidence

## Abstract

Light funneling determines how enhanced energy flows into subwavelength slits. In contrast to the previous research on oblique incident light, this study reveals that light funneling in the slits can be highly asymmetric, even at small angles. This mechanism is explained by polarized fields and charges, which are induced using Poynting vectors. It is shown that when light is obliquely incident to the slits perforated in a perfect electric conductor, asymmetrical fields and charges accumulate at the upper apex corners of the left (right) sides. When light is incident from the left (right) side, more (less) induced fields and charges accumulate in the left (right) slit corner so that the funneling width, area, and energy flow at the left (right) side increases (decreases).

## 1. Introduction

According to traditional electromagnetic (EM) theory, it is difficult for EM waves to penetrate a hole patterned in subwavelength nanostructures [1,2]. Transmission through a metal screen with subwavelength-sized holes, however, can be drastically increased. This is so-called extraordinary optical transmission [2,3,4,5,6,7,8,9,10,11,12,13]. To understand the physics underlying this phenomenon, Porto et al. [3] studied the transmission of two-dimensional gratings and found that the capability to control the free electrons on the metal surface was driven by the ability of the alternating incident electric field to collectively oscillate as surface plasmons. Takakura [4] performed an analysis of a single slit in a perfect electric conductor (PEC) and obtained the Fabry–Perot resonance wavelength. The detailed mechanisms and many different subwavelength structures have been studied [14,15,16,17,18,19] because of their importance for academic research. It also has numerous applications [20,21,22,23,24,25,26,27,28,29,30], such as in plasmonic lattices used as couplers to guide energy in an underneath subwavelength titanium dioxide layer, resulting in photonic crystal slabs [30].

As a near-surface effect, light funneling is the main mechanism [4,5,6,10,11,12] for the extraordinary optical transmission phenomenon, in addition to surface plasmonics [2,3,4,5,10,11], and Fabry–Pérot resonance [4,5,6,10,11,12]. The charges accumulate at an interrupted surface [31,32,33,34,35,36,37,38] as illuminated by the incident light and draw the wave energy into its vicinity due to diffraction effects. The interaction between the incident fields and the radiation fields generated by the accumulated charge causes the incident energy to flow into the enhanced opening. This effect is inherent to the metallic structure and does not require surface plasmon coupling. Thus, a slit in a film of the perfect conductor is capable of magnifying light transmission. When the slit width is smaller than half the wavelength of the incident field, such as in small holes or very narrow gaps, only the fundamental mode can exist in the narrow apertures. Astilean et al. [12] analyzed the energy flow around a single slit using Poynting vectors and found that the energy near the entrance of the slit is concentrated into the slit. García-Vidal et al. [39] addressed how the grating collects the incident energy. Zeng et al. [40] observed light funneling at the entrance of a hole. Zeng et al. [41] studied light funneling into subwavelength grooves in metal-dielectric structures. Jazayeri et al. [42] studied all-dielectric structures for trapping nanoparticles via light funneling and nanofocusing. Kamalieva et al. [43] fabricated silicon nanostructures for application in photonics. Li et al. [6] studied the funneling profile of the enhanced transmission of normal incident light through a subwavelength slit in a perfect electric conductor. The horizontal boundary of the funneling profile was defined by the time-averaged Poynting vector in the *x* direction to explicitly indicate the area where the energy could be collected by the slit. They concluded that the funneling areas of both sides were symmetric. Pardo et al. [5] explained light funneling of an oblique incident light as a consequence of magnetoelectric interference (MEI) [4,44,45,46] effect. They considered an optimized device at the resonance wavelength so that no wave was reflected and the funnel widths of both sides were almost symmetrical even at large incident angles, such as 30° and 50°. The streamlines of their total Poynting vector consisted of three components: the Poynting vector of the incident field *S_i_*, the Poynting vector of the evanescent field *S_e_*, and the Poynting vector *S_ei_* of the interference between the incident wave and the evanescent field. When the light entered at an angle, the symmetry was broken, however *S_i_* did not funnel. The evanescent field *S_e_* was symmetrical, indicating that the electric field *E_e_* and magnetic field *H_e_* were symmetrical as well. The interference *S_ei_*, or the MEI, was what caused the funneling. While the region away from the slits was not symmetric due to the already nonsymmetric incident fields, the region at the immediate entrance of the slit was symmetric. While *S_e_* did not channel energy into the slit, it modified the funneling shape and overall profile to become almost symmetrical. In contrast, we found that light funneling is highly asymmetric, even at much smaller incident angles.

This paper investigates the interaction mechanism between an obliquely incident [47,48] *p*-polarized light and subwavelength slits pierced in a perfect electric conductor (PEC) film to understand the angle of incidence effect on funneling and its profile [49,50]. For simplicity, the Fabry–Perot effect was not considered. The wave dynamics, especially the EM fields and Poynting vectors, were simulated using the finite-difference time-domain (FDTD) method. When light was obliquely incident on the PEC, the Poynting vectors of light fields arriving at the layer at half a wavelength away from the metallic surface bent to a normal direction like those of normal incident. Interestingly, at a quarter wavelength away, the Poynting vectors bent again to form a funnel and the shape was asymmetric. We observed that the asymmetric polarized fields and charges accumulated at the upper apex corners. When light was incident from the left side, more (less) induced fields and charges accumulated in the left (right) slit corner so that the funneling width, area, and energy flow at the left (right) side increased (decreased). The physical mechanisms will be interpreted in detail.

The funneling area and the time-averaged Poynting vectors [51] of light funneling into the subwavelength slits under different incident light angles will be presented. The electric field amplitude and the magnitude of Poynting vectors of the *y* component at the PEC interface will be investigated.

## 2. Materials and Methods

We simulated a *p*-polarized plane wave with wavelength λ = 640 nm illuminating a PEC film perforated with subwavelength slits. The source was positioned one wavelength in the *y* direction above the PEC film at various incident angles *θ_i_*, and its magnetic field in the *z* direction, *H*_z_, was directed out of the page. A Drude model was employed with a damping coefficient of zero and a sufficiently high plasma frequency of *ω_p_* = 1.0 × 10^30^ Hz to approximate the property of PEC because conductivity is proportional to the plasma frequency squared [52]. The slit periodicity was λ/2, each slit width was 80 nm, and the film had a thickness of *h* = 13 μm to avoid wave reflection from the slit exit. The simulation system size was 26 × 14 μm^2^ and the cell size was 5 nm.

The directional energy flux of the EM field was studied using streamlines by normalizing time-averaged Poynting vectors:(1)$$\langle S_{x}\left(x,y\right)\rangle =\frac{1}{S_{0}}\frac{1}{T}\int_{t_{a}}^{t_{b}}S_{x}\left(x,y,z\right)dt$$
(2)$$\langle S_{y}\left(x,y\right)\rangle =\frac{1}{S_{0}}\frac{1}{T}\int_{t_{a}}^{t_{b}}S_{y}\left(x,y,z\right)dt$$
where *t_a_* is the initial time, *t_b_* = *t_a_* + *T* is the final time of the average over one period, *T* is the period of the incident wave, and *S*_0_ is ½ the magnitude of the Poynting vector normalized to the incident electric and magnetic field amplitudes. For an obliquely incident light reflected by a perfect conductor similar to our case, the normalized time-averaged Poynting vector *S_avg_* has an analytical solution [53]:(3)Savg=Re((Ei+Er)×(Hi+Hr)*)2=x^kωμ2 cos2(kycosθi)sinθi
where *E_i_* is the incident electric field, *E_r_* is the reflected electric field, *H_i_* is the incident magnetic field, *H_r_* is the reflected magnetic field, *k* is the wave number, *ω* is the angular frequency, and *μ* is the permittivity in the vacuum. The vector *S_avg_* is at a minimum at $y=\left(2n+1\right)\lambda /(4\cos\theta_{i}),$ where *n* is a natural number, while *S_avg_* is at its maximum at $y=n\lambda /\left(2\cos\theta_{i}\right)$.

## 3. Results

### 3.1. Asymmetric Funneling at Oblique Incidence

The simulation results of time-averaged Poynting vectors fields of a *p*-polarized plane wave illuminating a PEC film at different angles are shown in Figure 1. The colored contours represent where the Poynting vectors *S*_x_ ≠ 0. The funneling boundary is the region between the grey and colored contours where the wave flows into the slit. The boundary enclosing the colored contours is the funnel area.

At normal incidence, the time-averaged Poynting vectors for incident energy flow were equidistant as they flowed along the propagation direction as indicated in Figure 1a. As the energy flowed into the slits, the funneling sizes near the slits were symmetrical and equal in magnitude. When the fields penetrated into the metal, the field distribution was also symmetrical and equal in magnitude. In contrast, at small incident angles 5° and 10° to the normal, the penetrated field distributions inside the metal became highly asymmetrical as shown in Figure 1b,c. The funneling size and the magnitude to the left of each slit was increased while the size to the right was decreased. This was more evident as the angle of incidence was increased. For both cases, the energy flow entered at an angle until *y* ≈ 200 nm; it was turned to downward only in the case of normal incidence until *y* ≈ 125 nm; and it was then returned back to its original angle of incidence again before funneling into the slits. We found this to be an interesting continuation of normal incidence; as observed in the second region, (200 nm > *y* > 125 nm) would be expected. For example, when the angle of the incident wave was 10°, the field had a minima *S_avg_* = 0 at *y* ≈ 162 nm and 487 nm and a maxima *S_avg_* ≈ 0.35 at *y* ≈ 0 nm, 325 nm, and 650 nm. The energy flows were bent from *y* = 325 nm to 162 nm, straight at *y* = 162 nm, and bent again from *y* = 162 nm to 0 nm.

The symmetry of the funnel boundary was manipulated when the angle of the incident light was changed. At incident angles 0°, 5°, and 10°, the width of the left (right) boundary was 160 nm (160 nm), 198 nm (122 nm), and 230 nm (90 nm), respectively. The sum of the funnel widths was λ/2 = 320 nm due to the periodicity of the slits. When the left (right) width of the boundary became larger (smaller), the left (right) height of the boundary increased (decreased) as the incident angle simultaneously increased. As shown in Figure 2, the left (right) funnel area for incident angles 0°, 5°, and 10° was 1.56 × 10^4^ (1.56 × 10^4^) nm^2^, 2.29 × 10^4^ (1.02 × 10^4^) nm^2^, and 3.08 × 10^4^ (0.55 × 10^4^) nm^2^, respectively.

We found that more (less) EM wave energy flow redirected inside the left (right) funnel area under oblique incidence. An interesting feature was that, as the symmetry of the funnel widths was broken, the left (right) funnel width was increased (decreased) as the energy flow increased (decreased). Thus, the transmitted energy was positively correlated to the funnel area.

### 3.2. Mechanism of Asymmetrical Funneling

The light funneling effect is an interaction between the incident fields and charges induced around the corners [54]. As shown in Figure 1, the fields penetrated into the metal at the corners of the slits and we could clearly observe the penetrating field distributions. When the angle of incidence was normal to the PEC, the penetrating fields at both corners were symmetrical. As the angle was increased from 0° to 5° to 10°, the penetrating field distribution became nonsymmetrical; the penetrating fields in the left corner became greater than that of the right corner. The penetrating fields at the corners were linearly proportional to the accumulated induced charges as they had to be self-consistent to satisfy Maxwell’s equations. Thus, the field distribution corresponded to charge distributions at the corners.

When the incident wave reached the PEC at *y* = 0, its propagation path was changed. While the *y* component of the electric field changed, the magnetic field remained constant. As the angle of incidence was increased, the amplitude of their instant normalized electric field at the left (right) corner also increased (decreased), as shown in Figure 3, when the magnetic field at the left corner reached its positive maximum. The increase (decrease) of the vertical electric field *E_y_* at the left (right) corner as indicated in Figure 3a changed the energy flow at the interface. This affected the funnel boundary as the horizontal boundary of the funneling profile was defined by the Poynting vector for the *x* component *S_x_*, which is proportional to *E_y_* × *H_z_*. The phase of the vertical electric field *E_y_* in the left (right) side decided the *x* component of the Poynting vector to be positive (negative). As the angle of incidence increased, the left (right) side boundary of the light funnel enlarged (decreased). The funnel width was determined by the electric field normal to the film.

The asymmetrical funneling effect was further examined quantitatively by studying the horizontal electric field *E_x_* near the slit entrance, as shown in Figure 3b. The Poynting vector for the *y* component *S_y_* was proportional to −*E_x_* × *H_z_*. As the angle of incidence was increased, the amplitude of the horizontal electric field *E_x_* in the left (right) corner of the central subwavelength slit was increased (decreased) as well. The surface charges at the left corners accumulated. The edges further enhanced the local electric field, the Poynting vectors, and the funneling. Regarding the edge effect at the left corner, the horizontal electric field *E_x_* moved the charges on the surface to form the current towards the +*x* direction. The vertical electric field induced the current on the slit surface in the +*y* direction so that more positive charges accumulated to enhance the local electric field. At oblique incidence, the electric field at the right corner had a different phase, which should be considered for understanding the detailed mechanism.

Figure 4 shows the magnitude of the normalized Poynting vector of the *y* component |*Sy*| at the interface *y* = 0 for different incident angles. At normal incidence, the magnitude of the directional energy flux at both the corners of the center slit were |*S_y_*| = 2.87. When the angle of incidence was increased, e.g., 40°, |*S_y_*| was 4.36 (0.80) in the left (right) hot spot. The magnitude of |*S_y_*| was proportional to the magnetic field *H_z_* of the incident wave, and the sharpness at the edges was directly proportional to the number of accumulated charges. If the intensity of the spike increased, the amount of local energy funneled into the slit increased. On the other hand, the total energy fluxes, calculated by the integration over *x* at *t* = 3.3 *T*, were 146.20, 145.48, 143.62, 142.11, and 139.56 for angles of incidence 0°, 10°, 20°, 30°, and 40°, respectively. The total energy flux only slightly decreased when the incident angle was increased. This insensitivity to the incident light angle is academically interesting because the flux of light passing through an oblique plane is well-known to decrease significantly with the angle. In addition, this funneling feature is important for photonic applications.

## 4. Conclusions

We studied the funneling mechanism of oblique light incidence at the entrances of metallic subwavelength slits using FDTD simulations. In contrast to previous studies, we found that the funneling can be highly asymmetric for obliquely incident light, although the energy flow turns to downward only at normal incidence at a region near the surface. The mechanism is explained by induced fields, which correspond to polarization charges, using Poynting vectors. Regarding the dynamics of wave-charge interaction, accumulated fields and charges at the sharp edges of the slits caused the nearby light energy to flow into the slit, further enhancing light transmission. When light is incident obliquely from the left (right) side of the nano slit, asymmetric and more (less) polarized fields and charges are accumulated at the upper apex corner of the left (right) side, respectively, thus increasing (decreasing) the funnel width, area, and energy flow on the left (right) side. The total transmitted energy flux is insensitive to the incident light angle, in contrast to the light flux passing through an oblique plane. In addition to the academic significance for further understanding of the angle of incidence effect on light funneling, this study is useful for the development of subwavelength-scale plasmonic applications.

## Figures and Tables

**Figure 1 nanomaterials-13-00061-f001:**
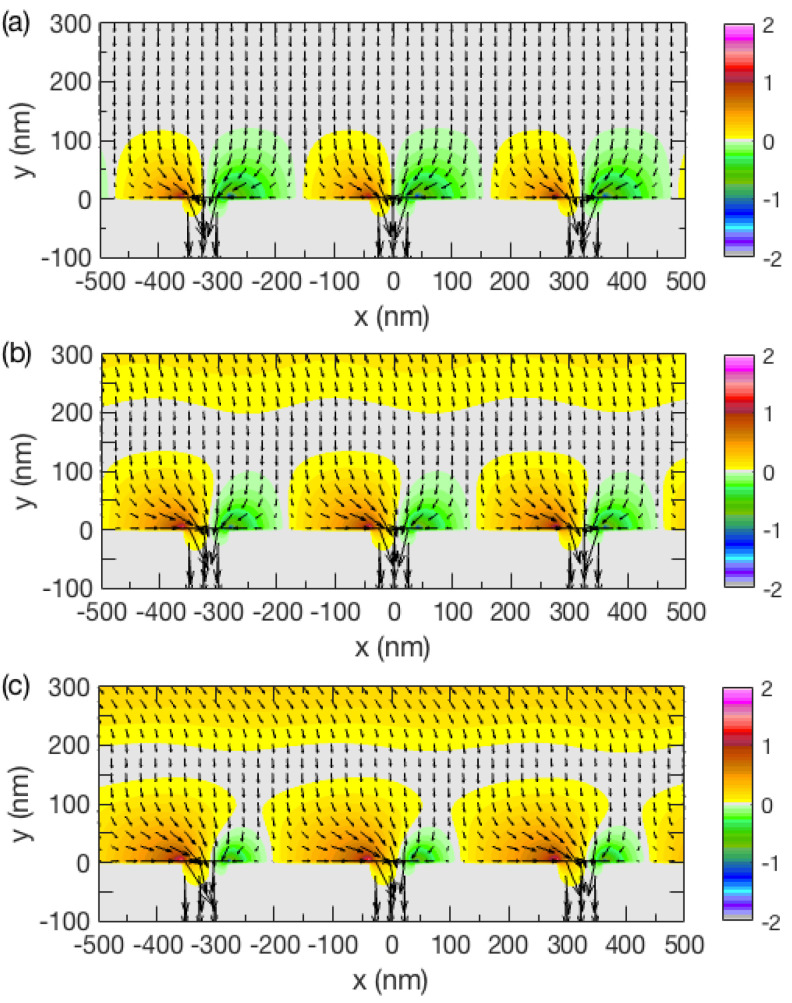
The time-averaged Poynting vector fields of light energy funneling at incident angle (**a**) 0°, (**b**) 5°, and (**c**) 10° into subwavelength grooves etched in a perfect electric conductor.

**Figure 2 nanomaterials-13-00061-f002:**
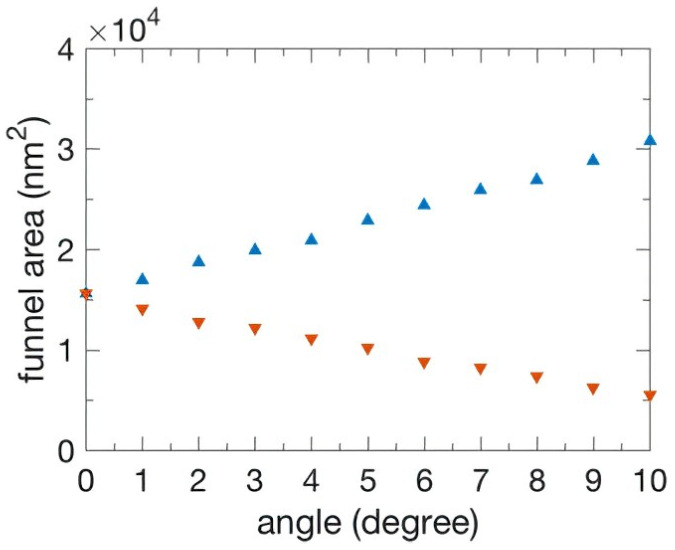
The relationship between funnel area and the angle of incidence. The left (right) funnel area is indicated by the blue upward-triangle (orange downward-triangle).

**Figure 3 nanomaterials-13-00061-f003:**
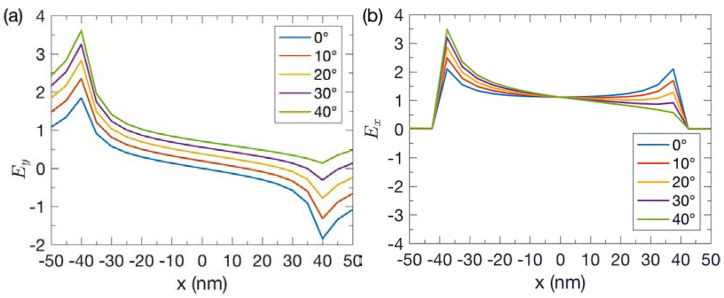
The amplitude of instant normalized electric fields (**a**) *E_y_* and (**b**) *E_x_* at the PEC vacuum-interface (*y* = 0) at various incident angles.

**Figure 4 nanomaterials-13-00061-f004:**
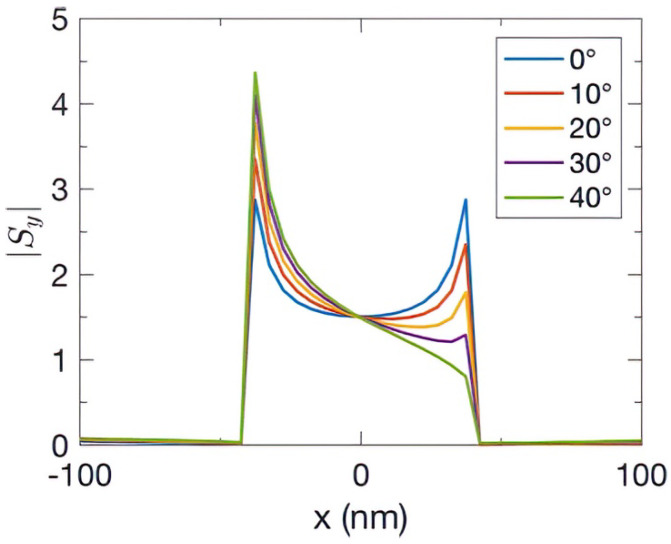
The magnitude of normalized Poynting vector of the *y* component |*Sy*| at the PEC interface, *y* = 0, at various incident angles, when the magnetic field at the left corner reaches its positive maximum.

## Data Availability

The authors confirm that the data supporting the findings of this study are available within this paper.

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
