# Peer review of "Funneling of Oblique Incident Light through Subwavelength Metallic Slits"

_nanomaterials, 2022, doi:10.3390/nano13010061_

Round 1

Reviewer 1 Report (New Reviewer)

The paper investigates the interaction mechanism between an obliquely incident p-polarized light and subwavelength slits pierced in a perfect electric conductor (PEC) film to understand the angle of incidence effect on funneling and its profile. I like the paper in its present form, and I found the results quite interesting. I encourage the authors to compare their results with previous calculations of plasmonic properties where small holes or very narrow gaps are used and point out the differences.

Author Response

Reviewer 2 Report (New Reviewer)

The authors in their research article ‘Funneling of oblique incident light through subwavelength metallic slits’ have explored theoretically using finite-difference timedomain analysis to demonstrate asymmetric funneling of light through subwavelengthslits. Using the Poynting vectors the authors have shown that for obliquely incident excitations to the slits perforated in a perfect electric conductor, asymmetrical fields and charges accumulate at the upper apex corners.

Comments:

The authors have analyzed the near-field effects regarding the asymmetric distribution and in the conclusion have also mentioned that the total transmitted energy flux is insensitive to the incident light angle, in contrast to the light flux passing through an oblique plane. However, since the manuscript is strictly confined to numerical simulations, it should be also beneficial for the readers if the authors can comment on the far-field
transmission characteristics of these slits upon normal and oblique incidence. This is important because most of the plasmonic applications involve the collection of far-field light. Thus, the importance of the current study can only be justified with FDTD calculated far-field data to show, whether this asymmetric distribution of the field affects the intensity profile or not. For the far-field property of metallic slits under normal and oblique incidences, the authors can refer to and introduce the following article:

[1] Sarkar, S., Gupta, V., Tsuda, T., Gour, J., Singh, A., Aftenieva, O., Steiner, A.M., Hoffmann, M., Kumar, S., Fery, A. and Joseph, J., 2021. Plasmonic Charge Transfers in Large‐Scale Metallic and Colloidal Photonic Crystal Slabs. Advanced Functional Materials, 31(19), p.2011099.

that involves plasmonic characteristics of 1D slits to highlight application-oriented analysis for an overall broad readership.

Author Response

This manuscript is a resubmission of an earlier submission. The following is a list of the peer review reports and author responses from that submission.

Round 1

Reviewer 1 Report

The revised manuscript is greatly improved with respect to the original version. In this revised form, I can recommend this manuscript for publication. There are still some sentences that are not completely clear, but I believe these issues can be dealt with at proof stage

Reviewer 2 Report

The manuscript entitled "Asymmetrical funneling of oblique incident light through subwavelength metallic slits" by A. Chen and his or her co-workers show the electromagnetic simulation of asymmetric funneling of a slit. Even though the result does not provide any faulty results under the considered simulation conditions, the detailed results do not carry the sounding conclusion of the term “asymmetric funneling of slits” owing to the imperfect slit and adjacent layer design and lack of simulation region of the FDTD. If two adjacent layers are included in the simulation region, asymmetric funneling cannot be obtained. Moreover, the reciprocity principle of electromagnetics without any time-reversal symmetry breaking – such as magnetization of the layer or external magnetic fields – cannot generate any asymmetric funneling of a slit sandwiched by two additional layers, which had been thoroughly discussed during the 1990s ~ the 2000s. Therefore, I do not recommend the publication of the current manuscript in Nanomaterials at all.

Reviewer 3 Report

The manuscript 'Asymmetrical funneling of oblique incident light through sub-wavelength metallic slits' is devoted numerical simulations of light fields passing through sub-wavelngth slits.

I have an ambivalent impression of this manuscript. On the one hand, this journal is not the most suitable for this manuscript: it is more related to electromagnetics. However, the material will be interesting for a broad scientific audience. On the other hand, the manuscript is of high quality; I reckon it is in the top 20% on the topic. Moreover, it seems like the manuscript was previously submitted to another journal and scientific discussion regarding light propagation and results presented in the manuscript were discussed. That is why I do not have critical comments regarding the topic.

My step-by-step analysis shows only two comments:

1. The authors provide a brilliant introductory part analysing many impressive achievements in the area. However, there is a sustainable trend in spreading ideas of plasmonics into nanophotonics, namely dielectric photonics. In this regard, I would recommend to analyse and mention several articles about this topic.

E.g., metal-dielectric structures:

1) Peng Zhu et al. Funneling light into subwavelength grooves in metal/dielectric multilayer films. Opt Express 21, 3595-3602, 2013.

And light funneling in dielectric/silicon and several interesting easy-to-perform methods of silicon nanoholes fabrication for light funneling investigation, e.g.:

1) Amir M. Jazayeri and Khashayar Mehrany, All-dielectric structure for trapping nanoparticles via light funneling and nanofocusing, JOSA B 34, 10, pp. 2179-2184 (2017).

2) Kamalieva, A.N., Toropov, N.A., Vartanyan, T.A. et al. Fabrication of Silicon Nanostructures for Application in Photonics. Semiconductors 52, 632–635 (2018). https://doi.org/10.1134/S1063782618050135

2. Formula (3) - typo.

I would recommend accepting this manuscript.